# Differential Item Functioning of the Mini-BESTest Balance Measure: A Rasch Analysis Study

**DOI:** 10.3390/ijerph20065166

**Published:** 2023-03-15

**Authors:** Antonio Caronni, Michela Picardi, Stefano Scarano, Peppino Tropea, Giulia Gilardone, Nadia Bolognini, Valentina Redaelli, Giuseppe Pintavalle, Evdoxia Aristidou, Paola Antoniotti, Massimo Corbo

**Affiliations:** 1IRCCS Istituto Auxologico Italiano, Department of Neurorehabilitation Sciences, Ospedale San Luca, 20149 Milano, Italy; 2Department of Neurorehabilitation Sciences, Casa di Cura Igea, 20144 Milano, Italy; 3Department of Biomedical Sciences for Health, Università Degli Studi di Milano, 20133 Milano, Italy; 4Department of Psychology and NeuroMI, University of Milano-Bicocca, 20126 Milano, Italy

**Keywords:** balance assessment, psychometrics, Rasch analysis, neurological rehabilitation, neurological balance impairment, falls risk assessment

## Abstract

The Mini-Balance Evaluation Systems Test (Mini-BESTest), a 14-item scale, has high content validity for balance assessment. This study further examines the construct validity of the Mini-BESTest with an emphasis on its measurement invariance. The Mini-BESTest was administered to 292 neurological patients in two sessions (before and after rehabilitation) and evaluated with the Rasch analysis (Many-Facet Rating Scale Model: persons, items, sessions). Categories’ order and fit to the model were assessed. Next, maps, dimensionality, and differential item functioning (DIF) were examined for construct validity evaluation. DIF was inspected for several clinically important variables, including session, diagnosis, and assistive devices. Mini-BESTest items had ordered categories and fitted the Rasch model. The item map did not flag severe construct underrepresentation. The dimensionality analysis showed that another variable extraneous to balance affected the score of a few items. However, this multidimensionality had only a modest impact on measures. Session did not cause DIF. DIF for assistive devices affected six items and caused a severe measurement artefact. The measurement artefact caused by DIF for diagnosis was negligible. The Mini-BESTest returns interval measures with robust construct validity and measurement invariance. However, caution should be used when comparing Mini-BESTest measures obtained with and without assistive devices.

## 1. Introduction

Balance and gait impairments due to neurological disease represent a main cause of increased risk of falling, a frequent and medically important problem [1,2]. Treatments are available to reduce fall risk. Notably, balance training reduces this risk [3], and enhanced balance is associated with a reduced risk of falling [4].

Therefore, measuring balance is clearly of the utmost importance. Balance measurement allows identifying patients with an increased falling risk (e.g., [5]). In addition, measuring balance allows for checking whether the patient is responding well to balance treatments such as therapeutic exercise. Finally, measuring balance before and after therapies allows for assessing, albeit indirectly, whether treatments have reduced the patient’s fall risk (e.g., [6]).

Many balance measures are available, both instrumental and from clinical tests. As instrumental balance measures, it is worth mentioning those from posturography [7,8,9], measures from the Timed Up and Go test [10,11], and measures of walking stability [12]. Among the balance scales, the Mini Balance Evaluation Systems Test (Mini-BESTest) [13] is likely one of the most used [14].

The Mini-BESTest consists of 14 items, assessing diverse balance facets such as static standing balance and balance during walking. This scale’s content validity [15] as a balance measure is high. In this regard, clinicians and scholars with expertise in balance and fall risk assessment and balance treatment recommend the Mini-BESTest for measuring balance construct [14].

A relatively young balance measure, the Mini-BESTest was developed in 2010 through the Rasch analysis [13].

The Rasch analysis of a questionnaire or a scale assesses whether the person’s answers to items comply with the requirements of the measurement model by Georg Rasch [16]. If this is the case, the questionnaire’s total ordinal score can be converted into an interval measure [17,18,19].

Questionnaires and tests complying with the Rasch model have two strengths. First, since their ordinal scores can be turned into interval measures, they provide measures like those of mathematics, physics, chemistry, and the like [17], i.e., “true” unidimensional measures. Second, a questionnaire passing the Rasch analysis has good construct validity [20,21].

The Mini-BESTest enjoys these two aforementioned strengths [13]. However, even if its good measurement properties have been confirmed [22], including by research groups independent from the scale developers [23], some studies pointed out some flaws of the scale [24] and the need for further assessment of its metric features [25]. In particular, the need for a deeper assessment of the Differential Item Functioning (DIF) of the Mini-BESTest items has been put forward [26].

DIF corrupts an item if persons belonging to different groups (e.g., patients affected by other diseases) but with the same amount of the measured variable (e.g., with the same balance ability level) score differently on that item [27,28,29]. When DIF is present for one or more items, the questionnaire score (e.g., the Mini-BESTest total score) can differ, although the quantity the questionnaire aims to measure (e.g., balance) is the same. Thus, DIF represents a real threat to measurement: differences between measures could just reflect a measurement artefact.

DIF should be preliminarily assessed whenever different measures (e.g., from other groups or different time points) are compared. For example, suppose the Mini-BESTest is used to compare the balance level of patients with different diseases such as stroke and peripheral neuropathy. In that case, it should be demonstrated first that its items function the same when used to measure balance in these two groups of patients.

Crucial in medicine is also verifying the invariance of the items functioning for the passing of time [30]. Most importantly, when a questionnaire or a scale is used as an outcome measure, the absence of DIF for treatment must be evaluated [31]. Showing the items’ stability before and after treatments increases confidence in drawing conclusions about treatment efficacy.

The primary aim of this work is to evaluate the measurement invariance of the Mini-BESTest items for several variables, including treatment and diagnosis. An in-depth analysis of the DIF of the Mini-BESTest items was run, and statistics from the Rasch analysis were used for this purpose.

## 2. Materials and Methods

The current study, part of a more extensive study to assess the risk of falling in neurological patients, recruited 292 persons from October 2018 to September 2020, all admitted to the inpatient rehabilitation unit of Casa di Cura del Policlinico (Milan, Italy) due to a neurological disability. All participants gave their written consent to the study, which was approved by the local ethics committee (Comitato Etico Milano Area 2, protocol code: 568_2018bis).

Participants were included if they were:Older than 18 years;Affected by one of the following five neurological conditions:
hemiparesis due to an ischaemic or haemorrhagic strokeperipheral neuropathy of the lower limbslumbar polyradiculopathy in spinal stenosisParkinson’s diseasevascular parkinsonisms;Able to walk without assistance from an operator;Able to move from sitting to standing without receiving assistance.

The exclusion criteria were:The simultaneous presence of two major neurological diseases (e.g., hemiparesis due to a stroke and Parkinson’s disease);The simultaneous presence of two conditions causing a gait or balance impairment (e.g., hemiparesis and severe heart failure).

All patients participated in physiotherapy and occupational therapy, one-on-one sessions (5 to 6 weeks; physiotherapy: two sessions/day, 45 min each, five days/week; occupational therapy: one session/day, 45 min each, three days/week). Details on the rehabilitation program, which followed evidence-based recommendations for rehabilitating patients at increased risk of falls (e.g., [32]), are given elsewhere [9].

### 2.1. Participant Assessment

Participants’ balance, disability, and locomotor ability were assessed at the beginning and end of their inpatient stay by administering the Mini-BESTest and the Functional Independence Measure (FIM) scale and recording the gait speed and the Timed Up and Go (TUG) test duration.

The Mini-BESTest [13,22] is a clinician-administered rating scale comprising 14 items, each scored on three categories (protocol available at www.bestest.us). Each item assesses the participant’s ability to complete a different balance task. For example, as representative items, in item 8 the ability to stand with feet together and with eyes closed on a foam surface is assessed, while item 11 investigates the ability to walk with head turns. Each item is scored 0, 1 or 2, and the total scale score ranges from 0 to 28, with higher scores indicating better balance.

Regarding the item scoring, the Mini-BESTest allows for assessing patients even when they need an assistive device to complete an item (www.bestest.us, accessed on 9 March 2023). However, when an assistive device is used, the item is scored one category lower. In addition, an item is scored 0 if the patient needs physical assistance. For the current study, patients were assessed without assistive devices whenever possible. However, if required, they were allowed to use their assistive device (most commonly a walker) for one or more items.

The FIM scale [33] returns criterion standard measures of disability. It consists of 18 items arranged into a motor (13 items) and a cognitive (5 items) domain. Each FIM item assesses a person’s ability to complete one of the primary activities of daily living, such as eating and transferring (from the motor domain) and interacting with others (from the cognitive domain). FIM items are scored on seven categories (from 1 to 7), with score 1 indicating that the person needs total assistance to complete a task and score 7 indicating that the task is completed autonomously. Therefore, the higher the total score, the better the patient’s condition.

The patient’s gait speed was measured using the 10 m walking test [34]. In this test, patients are asked to walk straight while the time spent travelling the central six meters of a 10 m linear trajectory is measured with a stopwatch.

The three meter variant of the TUG test [35] was performed. Participants were required to get out of a chair, walk three meters, turn around, walk back to the chair, and sit down. A traffic cone marked the turning point. The test started with a go signal from the experimenter. The time from this signal to the patient sitting down on the chair (i.e., the TUG duration) was measured with a stopwatch.

On admission and discharge, the 10 m walking test and TUG test were repeated five times each, and the median of these five trials was calculated. Participants were asked to complete both tests at their comfortable, safe speed.

All assessments were collected in a single session about one hour long (one session on admission and another on discharge). Gait and balance tests were usually administered in the following order: TUG test, 10 m walking test, and Mini-BESTest. Participants were allowed to rest between the different tests to avoid fatigue.

### 2.2. The Rasch Analysis of the Mini-BESTest Scale

The theoretical underpinnings of the Rasch analysis can be found in [17,18,19], while the technical details of the analysis run here can be found in [31,36,37]. Briefly, the Rasch analysis assesses whether the scores of questionnaire items comply with the Rasch measurement requirements of (i) ordered item categories, (ii) fit of data to the model of Rasch, (iii) unidimensionality, and (iv) absence of DIF.

If these requirements are verified, then unidimensional, generalizable interval measures can be obtained from test scores.

#### 2.2.1. Category Order

Regarding the Mini-BESTest, ordered categories mean that the item categories are numbered so that the higher the category’s numeral, the better the balance. Along with the order of the categories, the order of the Andrich thresholds is also commonly checked to assess the overall functioning of the categories. An Andrich threshold is a location along the latent variable (here, a balance level) at which adjacent categories (e.g., categories 0 and 1 of a Mini-BESTest item) are equally likely to be observed. According to some Authors [38], ordered Andrich thresholds represent the highest evidence that the categories work correctly.

#### 2.2.2. Fit to the Rasch Model

Mean square (MNSQ) and z-standardized (ZSTD) statistics have been used to assess the data–model fit. The MNSQ statistic is calculated from the model’s squared residuals and can be understood as an average of standardized residuals. MNSQ are then converted into the ZSTD statistics, z-scores with expectation 0.0 and variance 1.0.

MNSQ quantifies the size of the data departure from the model, and the ZSTD returns the statistical significance of this departure. MNSQ within the 0.5 to 1.5 range are commonly considered “productive for measurement”, while items with MNSQ > 2.0 “degrade the measurement” [39]. As customary with z-statistics, ZSTD values in the −1.96 to 1.96 range indicate that item responses are not significantly different from the model’s prediction. For the current analysis, both outlier-sensitive (outfit) and inlier-sensitive (infit) MNSQ and ZSTD statistics have been calculated (four data–model fit indices in total). The main difference between the outfit and infit statistics is that the latter is more robust to outliers. Poor outfit statistics could simply reflect some very unexpected answers to an item by a few respondents. In contrast, poor infit flags a more severe malfunctioning of the item in the sample of respondents. In statistical terms, it is worth mentioning that outfit MNSQ is the Pearson’s chi-square divided by its degrees of freedom. If the item scores fit the model of Rasch, the total questionnaire score can be turned into an interval measure. To this aim, when a Rasch-compliant test is developed, a raw score-to-measure conversion is commonly provided. With this, future test users can quickly transform the test total score into the interval measure as long as no items are missing. This score-to-measure transformation is given as a table (see Table 4 in [22]) or a plot (see Figures 4 and 5 in [33]).

#### 2.2.3. Assessing Unidimensionality: The PCA of the Model’s Residuals

The Rasch measurement model is unidimensional, meaning only one variable (here, balance) produces the item scores. Test unidimensionality is usually assessed with a principal component analysis (PCA) of the residuals of the Rasch model. The idea is straightforward: once the model expectation is “peeled off” from the item’s score, only random variation should be found in the residuals.

If the PCA highlights correlation patterns in the residuals, then one (or more) hidden variables affect the item scores in addition to the variable taken into account by the Rasch model. It is usually considered that if the PCA returns a principal component with eigenvalue > 2.0, an additional variable, hidden in the model’s residuals, drives the item scores (i.e., the test is multidimensional).

Strictly related to the concept of unidimensionality, there is that of local independence.

Local independence is the assumption that, once the latent variable is excluded, item responses are independent of each other. In simple terms, local independence implies that only the measured variable causes the items to co-vary [40]. Defined as such, the linkage between multidimensionality (i.e., the violation of unidimensionality) and local dependence (i.e., the violation of local independence) is clear.

For many psychometricians, the demonstration of unidimensionality grants that the local independence requirement has been satisfied [40]. However, not all scholars agree on this and consider local independence a more fundamental concept [40], with local independence including—but also going beyond—unidimensionality [41].

It has been proposed that local dependence would consist of trait dependence and response dependence [42].

Trait dependence is the local dependence caused by multidimensionality, i.e., additional traits that affect the item scores. The multidimensionality highlighted with the PCA of the model’s residuals would be trait dependence.

Response dependence is the local dependence caused by a loss of statistical independence between item scores. This dependence typically happens when success on an item (e.g., climbing three flights of stairs) implies success on another item (e.g., climbing one flight of stairs). Response dependence corresponds to one of the most common definitions of local dependence, i.e., local dependence occurs when responses to an item depend on the responses given to other items [43].

Even if trait and response dependence have been distinguished theoretically, these two types of local dependence are difficult to distinguish empirically [42]. In practice, the consequences of trait and response dependence on data are similar. For example, the correlation of item residuals is the hallmark of multidimensionality and local dependence [44]. Moreover, straightforward statistical tests are not available to easily distinguish between the two types of dependence [42].

On these bases, in the current work, only the violation of local independence due to multidimensionality will be assessed [41,43].

If multidimensionality is found, its effects on the measures are evaluated following the procedure by Smith [45]. First, the test items are split into “positive” and “negative” ones based on their loadings on the principal component. Then, respondents are measured with these two sets of items, and t-tests are calculated to compare if patient measures from the positive items are significantly different from the negative ones. Finally, the percentage of significant t-tests and the corresponding 95% confidence interval (CI; exact binomial test) are calculated. Multidimensionality is ignored if the percentage of patient measures that are significantly different when measured with positive and negative items is not substantially larger than 5%.

#### 2.2.4. Assessing Differential Item Functioning: Another Dimensionality Assessment

After dimensionality, DIF is assessed for a set of variables of clinical interest.

Regarding the Mini-BESTest, DIF tests whether scores for an item are the same in participants with the same balance level but belonging to different groups (e.g., males vs. females). If this is not the case, a variable distinct from balance affects the item’s score (e.g., gender).

DIF is just another facet of multidimensionality by this definition. However, it is common to consider the PCA of the model’s residuals as the test of dimensionality assessment and the DIF as a separate analysis.

The following six variables were tested for DIF:Session (before vs. after rehabilitation);Gender (males vs. females);Age (adults: <65 years; older adults: ≥65 years and <80 years; very older adults: ≥80 years);Presence of cognitive impairment (present vs. absent);Diagnosis (stroke vs. peripheral neuropathy);Use of an assistive device during testing (yes vs. no).

As the Introduction reports, DIF should be preliminarily assessed whenever measures from different groups of persons or time points are contrasted. Clinically important variables were selected for the DIF analysis run here. For example, it is paramount in the clinic to compare balance before and after rehabilitation and in patients with different neurological conditions.

Patients scoring < 35 (out of 35) on the cognitive domain of the FIM scale were considered cognitively impaired. Conversely, entirely in line with the FIM scale instructions, patients scoring 35/35 were supposed to suffer no cognitive impairment [31]. Persons scoring 35/35, i.e., scoring 7/7 in the five cognitive items, have no communication impairment, memory problem, or difficulty in problem-solving or social interaction. These persons can communicate and discuss complex cognitive problems such as consulting their physicians about their medicines and treatments [46].

As specified above, the Mini-BESTest allows using assistive devices during testing. For the DIF analysis, the Mini-BESTest scales were classified as completed without assistive devices or using an assistive device in one or more items.

For the current study, a Mini-BESTest item was considered corrupted by DIF if the item’s difficulty was significantly different (*p* < 0.01) in the two groups of respondents, and this difference was >0.5 logits.

DIF was assessed according to Linacre [47]. First, for each DIF variable tested here, the Mini-BESTest items were calibrated in the two groups of participants and the item difficulties (and their standard errors) were obtained. Next, the DIF significance was calculated from the t-statistic (i.e., the item difficulty difference/joint standard error) with joint degrees of freedom computed according to Welch-Satterthwaite. Finally, as recommended [48], the top-down purification procedure was used to identify those items affected by real DIF [49,50].

Different solutions can be put in place in the case it is found that one or more items are corrupted by DIF. Similar to multidimensionality, DIF can also be overridden if it is shown to have no major consequences on measures; i.e., if the measurement artefact it causes is minor.

In the current study, the artefact caused by DIF was quantified by comparing the score-to-measure transformations specific to the two DIF groups [51]. The next steps were followed.

First, items affected by DIF have been split according to the “split items” procedure [52]. For example, consider a test with one item (say item 2) more difficult for group A than group B respondents. Item 2 is thus affected by DIF. With the split items procedure, two virtual items are derived from the item with DIF. In the example reported above, these virtual items would be: “Item 2—group A” and “Item 2—group B”. In the dataset, “Item 2—group A” reports the patient scores belonging to group A and missing values for patients from group B. For “Item 2—group B”, it is vice versa. Next, a new analysis is run on the database with the split items (the original item 2 is discarded), and the item difficulties (split items included) are obtained. Finally, these calibrations are used to arrange two score-to-measure transformations, one for measuring group A and the second for group B patients.

It should be stressed here that the split items procedure solves the measurement artefact caused by DIF. The group-specific score-to-measure transformations from this procedure return unbiased measures that are safely comparable in the two groups of respondents (groups A and B in the example).

Consider now two respondents, one from group A and the other from group B, totaling the same test score. If DIF is ignored and the “overall” score-to-measure table is used to measure both, their measure would be the same; therefore, the group A participant measure would be underestimated. Conversely, if group-specific score-to-measure transformations are used, the group A participant would correctly measure higher than the group B participants.

However, is an essential difference between the two patients missed if the overall score-to-measure transformation is used to measure both? Is this error tolerable? In other words, is there a real need to use group-specific score-to-measure transformations? In the Rasch world, a difference should be at least >0.5 logits to matter [53].

By comparing score-by-score the two group-specific score-to-measure transformations, one can quickly check throughout the scale range whether the difference between the two measures is >0.5 logits. If this is not the case, the measurement artefact caused by DIF can likely be ignored.

#### 2.2.5. Person Measure Reliability

As the last Rasch analysis steps, measure reliability and the questionnaire’s maps are evaluated.

The Rasch analysis returns the persons’ measures and their standard error. Hence, it is possible to calculate a person’s reliability index, which entirely agrees with the classical definition of reliability (i.e., the ratio of the “true” variance to the variance of the observed measures) and is analogous to Cronbach’s alpha [54].

The number of “strata”, i.e., the number of measure levels significantly different at a single subject level, is easily calculated from the persons’ reliability. It is recommended that persons’ reliability is at least 0.8 so that the test can distinguish between three strata [55]. For example, a test with these features could track a patient’s improvement from severe to moderate and eventually mild impairment. Conversely, a test with two strata can only follow large modifications in a person’s ability, such as a patient improving from a severe to a favorable condition.

#### 2.2.6. The Facet maps

The item and person maps graphically display the item calibration and person measures along the line representing the construct of interest (balance, in this case). This simultaneous representation makes it immediately apparent whether the scale is well-targeted to the participant sample or whether it is affected by a ceiling or a floor effect. Moreover, the item map is a valuable tool for further assessing a questionnaire’s construct validity. For example, it allows one to quickly check whether the test items well probe the construct of interest or whether it is under-represented [21].

#### 2.2.7. Which Model of the Rasch Family?

The Many-Facet model [56] (Rating Scale variant) was used for the current analysis. The “Rasch model” represents a family of measurement models, spanning from the original one for the analysis of dichotomous items to the Many-Facet model, one of its most recent additions. The Many-Facet Rating Scale model returns Rasch measures from articulated measurement situations in which persons interact with polytomous items and a third facet (e.g., occasions). In statistical terms, this model provides a convenient way to run the Rasch analysis when there are repeated measurements, which is why it has been used here. As explained above, participants were assessed twice (on admission and on discharge). Hence, two Mini-BESTest scales were collected for each participant (584 in total). A three-facet model was used: participants, items, and sessions.

Facets version 3.84.0 was used for the primary analysis, and Winsteps version 5.2.5.2 for calculating the PCA of the model’s residuals. R version 4.2.0 was used for the remaining statistics and graphics.

## 3. Results

### 3.1. Patients’ Clinical Features

Patient features are detailed in Table 1.

Most patients were males (56.2%), were more than 65 years (79.8%), and had hemiparesis secondary to stroke (53.1%). Peripheral neuropathy was the second most common diagnosis (21.2%).

At admission, neurological patients suffered a moderate disability according to the motor domain of the FIM scale (median score: 66; 1–3Q: 50–75).

The median value of the sample’s FIM cognitive domain was 33 and 34 on admission and on discharge, respectively, indicating minimal to no cognitive impairment in many patients. About 43% of the patients scored 35 (i.e., full score) on the discharge assessment.

The total score of the Mini-BESTest scale (14; 9–18), the gait speed (0.72 m/s; 0.54–0.98), and the TUG duration (17.7 s; 13.2–25.5) are consistent in indicating that the patients suffered moderate balance and gait impairments, which significantly improved at discharge (for all three measures, Wilcoxon signed rank test with continuity correction: V ≥ 33,040; *p* < 0.001).

### 3.2. Rasch Analysis—First Run: Whole Sample Analysis

The first analysis of the entire sample of patients showed that Mini-BESTest items had ordered categories and Andrich thresholds, indicating that the item categories worked as intended.

All items correctly fit the model of Rasch, even if the infit of item 7 (feet together, eyes open, firm surface) was borderline (MNSQ: 1.42; ZSTD: 3.6).

Regarding the dimensionality analysis, the eigenvalue of the first principal component from the PCA of the model’s residuals was >2 (admission: 2.6; discharge: 2.4). In both analyses, items 4 (compensatory stepping—forward), 5 (compensatory stepping—backward), and 6 (compensatory stepping—lateral) had a large, positive loading (≥0.69) on the first component.

When single participants’ measures from items with positive loadings were compared with those from negative loading items, measures significantly different were found in 8.2% (95% CI: 5.3–12.0%) of the sample on admission and 8.6% (95% CI: 5.6–12.4%) on discharge.

At the sample level, on admission, the mean measure from positive and negative loading items was −0.04 (SD = 2.10) and −0.17 (SD = 1.94) logits, respectively. On discharge, the sample mean measures from the opposite cluster of items were 0.82 (SD = 2.32) and 0.91 (SD = 1.61) logits.

Cohen’s d of these differences, i.e., the effect size, was 0.06 (95% CI: −0.05 to 0.17) on admission and −0.04 (95% CI: −0.15 to 0.07) on discharge, indicating that at the sample level the measurement artefact caused by multidimensionality is negligible.

The most striking finding from the DIF analysis was that six items had DIF for assistive device use (see next section for the complete DIF analysis). More specifically, items 1 (sit to stand), 4 (compensatory stepping—forward), 6 (compensatory stepping—lateral), 10 (walking with speed changes), 12 (walking with pivot turns), and 13 (stepping over obstacles) were all more difficult in patients using an assistive device than those not using an assistive device.

Figure 1A shows the score-to-measure curves of the Mini-BESTest for patients who completed the tests with an assistive device and those who did not. The central portion of the curve for assistive device users is shifted upward (Figure 1A, leftmost and middle panels). In addition, for Mini-BESTest total scores >16, the difference between the measures from the two transformation curves is >0.5 logit (Figure 1A rightmost panel). Overall, the score-to-measure plots from the split items procedure indicate that the artefact caused by the DIF for assistive device use on the Mini-BESTest measure is not negligible.

See Appendix A for additional analyses.

### 3.3. Rasch Analysis—Second Run: Patients Using an Assistive Device Excluded

A second analysis was run on the sub-sample of patients completing the tests without the assistive device (223 participants who did not use assistive devices in both sessions and 29 participants who used an assistive device on admission but not on discharge; 475 tests in total).

Similar to the previous analysis, the Mini-BESTest items had ordered categories and Andrich thresholds (Table 2).

All the Mini-BESTest items properly fit the Rasch model (infit MNSQ range: 0.74–1.30; outfit MNSQ range: 0.58–1.23; Table 3), and item 7 fit much improved.

The PCA results were superimposable to that of the previous analysis. The eigenvalue of the first principal component was 2.7 on admission and 2.5 on discharge. As before, items 4, 5, and 6 had a large and positive loading on this component. Again, measures from positive and negative items differed significantly in 8.7% (95% CI: 5.6–12.9%) and 10.3% (95% CI: 6.9–14.8%) of participants on admission and discharge, respectively.

However, in this analysis, the difference in measures from the two opposite clusters of items was also negligible at the sample level (Cohen’s d, admission: −0.04, 95% CI: −0.17 to 0.09; discharge: −0.02, 95% CI: −0.15 to 0.12).

No DIF was found for session (before vs. after rehabilitation), age, and gender.

Item 14 (TUG with a dual task) showed DIF for cognitive impairment (Table 4) was more difficult for patients with cognitive impairment than for those without.

Moreover, four items were affected by DIF for diagnosis. Item 7 (standing feet together with the eyes open on a firm surface) was more difficult for PNLL than stroke patients. On the contrary, items 11 (walking with head turns), 12 (walking with pivot turns), and 14 (TUG with a dual task) were more difficult for stroke patients.

Contrary to the DIF analysis for assistive device use, the artefact caused by DIF for cognitive impairment and DIF for diagnosis on the Mini-BESTest measures was small.

The split-to-measure curves for patients with and without cognitive impairment are virtually superimposed (Figure 1B). Regarding the DIF for diagnosis, the two score-to-measure curves are slightly apart (Figure 1C). However, except for score 0, the differences in measures for the Mini-BESTest total scores are below 0.5 logits (Figure 1C, rightmost panel).

The sign of the DIF for item 7 was opposite to items 11, 12, and 14. Hence, the measurement artefact caused by items 11, 12, and 14 was partly cancelled out by the one caused by item 7. Therefore, a control analysis was run after removing item 7, which confirmed that the measurement artefact caused by the DIF for items 11, 12, and 14 was below the 0.5 logits threshold (see Appendix A).

The Mini-BESTest maps are shown in Figure 2.

The lowest balance level is flagged by the item easiest to succeed at—item 7 (standing with feet together and eyes open on a firm surface; about −3.5 logits). In contrast, the most challenging balance task is standing on one leg (item 3), the item with the highest calibration (almost two logits).

The test targeting is relatively good, with the person measures (mean: 0.78 logits; SD: 1.31 logits) slightly above the items’ mean calibration (i.e., conventionally 0 logits). In addition, the scale had virtually no ceiling or floor effect, since only one participant scored 0 (i.e., the Mini-BESTest minimum total raw score) on the admission assessment, and three participants totaled 28 (i.e., the scale’s maximum score) on discharge.

Overall, the item thresholds are well distributed along the balance continuum (Figure 2B), as confirmed by the median distance between consecutive thresholds, which was 0.12 logits (1st to 3rd quartile: 0.06 to 0.325 logits). However, thresholds are sparse at the map’s lower end, where three gaps >0.5 logits exist between consecutive thresholds.

The Mini-BESTest maps also highlight the different calibrations of the two sessions, with the discharge session measuring lower (−0.44 logits) than the admission one (0.44 logits; Figure 2D).

Person (sample) reliability was high (0.91), and the Mini-BEStest can distinguish at least four balance levels (number of strata: 4.64).

Finally, the score-to-measure conversion table is reported in Table 5.

## 4. Discussion

In the current work, the Rasch analysis was used to assess the metric properties of the Mini-BESTest scale, a criterion standard for balance measurement in neurological diseases [14]. It is shown here that the Mini-BESTest total scores can be turned into interval balance measures, since its items fit the Rasch model. Furthermore, these measures have high reliability and satisfactory construct validity.

Overall, the present results:Show that the multidimensionality of the Mini-BESTest is relatively weak;Support the use of the Mini-BESTest as an outcome measure for assessing improvement after treatment for balance impairment in patients with neurological disorders, since no DIF was found for treatment;Support the use of the Mini-BESTest measures for comparing balance in different diseases since, even if DIF for diagnosis was found, this was modest;Show that the artefact caused by the DIF for assistive devices on the Mini-BESTest’s balance measures should not be ignored.

A main strength of the present study, with respect to previous ones, is the examination of the Mini-BESTest measurement invariance by the DIF assessment; in particular, the DIF for treatment has been investigated. Showing that a questionnaire or a scale is free of DIF for treatment is of utmost clinical importance, since this implies that the questionnaire can be safely used to compare patients before and after treatment and thus decide on the treatment’s efficacy. In addition, other clinically essential variables—diagnosis in the first place and assistive devices—were also evaluated for DIF.

The ongoing reexamining of the psychometric features of widely used measures should not be underestimated [57], and regarding the Mini-BESTest, further psychometric studies have been recommended [25]. The analysis reported here comes from a research group independent of the scale developers and offers data from a new sample of neurological patients, therefore representing a further check of the scale.

### 4.1. Assessing Questionnaire Construct Validity with the Rasch Analysis

The Rasch analysis is not just about checking whether data fit the model of Rasch. On the contrary, this analysis offers tools such as the item map, the dimensionality assessment, and the study of DIF to evaluate the construct validity of a questionnaire.

According to the COnsensus-based Standards for the selection of health Measurement Instruments (COSMIN) initiative, questionnaire scores have construct validity if they are consistent with hypotheses about the construct to be measured [58]. For example, verifying assumptions about the questionnaire’s dimensionality is crucial in evaluating the questionnaire’s structural validity, a facet of construct validity [59]. Examples of these hypotheses in the Rasch measurement framework are that item scores are unidimensional and free of DIF.

#### 4.1.1. Dimensionality Analysis of the Mini-BESTest

The PCA of the model’s residuals showed that a second variable drives the scores of the Mini-BESTest items in addition to balance. This result was consistently found in the analysis of the whole participants’ sample (including patients using an assistive device) as well as in the subsample of patients not using assistive devices. In addition, multidimensionality was found in admission and discharge data. However, the current analysis also showed that the multidimensionality of the Mini-BESTest was relatively weak.

First, a single principal component with an eigenvalue >2 was found.

Second, in the different conditions, the eigenvalue of this principal component was between 2.4 and 2.7. Therefore, this hidden variable affected the score of three (out of 14) items at most.

The items with the largest loadings on this component were items 4, 5, and 6, all of them evaluating the ability to recover balance after leaning against the examiner. It is not surprising that precisely these three items are affected by multidimensionality if a look at their content is taken. At least three variables interact when these items are tested: the ability to recover balance, the examinee’s confidence in leaning against the examiner, and the examiner’s confidence.

In this scenario, it can be speculated that the examiner’s confidence could be the hidden variable: the lower the confidence in the patient succeeding at the item, the lower the amount of leaning requested. Hence, the easier the task for the examinee, who can recover balance with a single step, thus totaling the maximum item score.

Items are commonly dropped from the questionnaire or split into subdomains (which work as different questionnaires) when multidimensionality is found. However, these solutions should not be used for the Mini-BESTest, even if multidimensionality has been found.

Splitting the questionnaire into two domains would create a questionnaire consisting of just three items (nine categories in total), whose reliability would be dramatically low. In addition, removing these three items would likely resolve multidimensionality, but this would affect the content validity of the Mini-BESTest. Testing the ability to recover balance is fundamental in balance assessment of neurological patients [8].

It should also be noted that even if—when measured with positive and negative loading items—participants’ measures are significantly different in more than 5% of the sample, this percentage remains low, ranging from 8.2% to 10.3% in the other analyses.

Finally, when interpreted in terms of effect size, the effects of multidimensionality on Mini-BESTest measures are negligible at the sample level, as pointed out by Cohen’s d values.

#### 4.1.2. Differential Item Functioning of the Mini-BESTest

DIF is another way to assess a questionnaire’s dimensionality.

When an item is affected by DIF, it means that the DIF variable (e.g., the assistive device use) affects the item scores independently of the variable the questionnaire is aimed to measure (e.g., balance). Therefore, being another dimensionality test, DIF can be considered another construct validity test.

The DIF analysis of the Mini-BESTest showed that assistive devices similarly affected items 1, 4, 6, 10, 12, and 13. All six of these items were more difficult in patients using an assistive device than in those not using an assistive device, a finding somehow contrary to what one might expect, since assistive devices are used to make a task more accessible.

The score-to-measure curves also highlighted that the artefact caused by this DIF on the Mini-BESTest balance measures could not be ignored. Consequently, caution is needed when comparing the balance measure of a patient who completed the Mini-BESTest without assistive devices with that of a patient who used an assistive device for one or more items. Caution is also needed when the same patient used the assistive device in the first assessment (e.g., before rehabilitation) but they did not use it anymore at the subsequent evaluation (e.g., at the end of rehabilitation).

The Mini-BESTest allows using an assistive device during testing. The test instructions specify that if an assistive device is used for an item, that item should be scored one category lower. In other words, participants with an assistive device can only score 0 or 1 on the item they completed with the device. The present DIF analysis indicates that this correction (i.e., “remove 1 point”) for using an assistive device is too severe. Also, to our knowledge, this procedure has not been experimentally validated.

Regarding items 1 (sit to stand), 10 (walking with speed changes), and 12 (walking with pivot turns), these are balance tasks that can be efficiently completed with an assistive device. On the contrary, item 13 (stepping over obstacles) cannot be completed with a walker, the most-used assistive device in the sample, leading the patient to score 0 on this item.

The same happened for items 4 and 6, which can hardly be administered if a patient needs an assistive device for standing. Again, according to the scale instructions, the patient scored 0—the lowest balance category. It is noteworthy that, according to the PCA, these items were affected by multidimensionality. Accordingly, it can be further speculated that the examiner’s confidence could also cause the DIF for assistive devices. If the confidence was the lowest, the item was not administered at all.

Testing DIF for diseases is essential with generic tests and questionnaires. A questionnaire should have no DIF for diagnosis if one is interested in comparing the measured variable in clinical groups with different diagnoses. For example, for a comparison such as: *“patient A, who had a stroke, has better balance than patient B, who is affected by a peripheral neuropathy”*, it must have been preliminarily demonstrated that the balance measure works equally in stroke and peripheral neuropathy.

The Mini-BESTest is a disease-free balance measure, since it was developed initially on a sample of neurological patients with various diagnoses [13]. Contrary to disease-specific measures (e.g., [60]), generic measures have the strength of allowing the comparison of different diagnostic groups. However, one must first test experimentally if these generic measures are indeed invariant, i.e., DIF-free, across diagnoses [61,62].

Even if it is shown here that four items were affected by DIF for diagnosis, it is also demonstrated that the measurement artefact caused by this type of DIF has modest consequences on the Mini-BESTest balance measures. Hence, the Mini-BESTest can be considered sufficiently invariant across the two neurological groups tested here.

The current work compared the score-to-measure curves from the two DIF groups to assess DIF’s effects on questionnaire measures. This method, used for the first time for the Mini-BESTest, is a variation of the “equated scores” method [51,63,64]. Briefly, scores from one group are adjusted (i.e., increased or decreased) to make them comparable to those of the other group, and the difference between the two equated scores is eventually evaluated [63].

Another procedure often used to examine the effects of DIF on the whole questionnaire has been suggested by Tennant and Pallant [65]. Similar to what was done here to investigate multidimensionality, persons are measured with a set of items with no DIF (pure items) and with the whole questionnaire. DIF effects on measures are usually ignored if the proportion of patients measuring differently with the two analyses is <5%.

Comparing the score-to-measure curves has, however, two advantages. First, it allows for assessing measure invariance throughout the scale, while the pure items technique strictly depends on the measure distribution of the sample who participated in the study. Second, it shifts the attention from statistical significance to effect size.

The method based on comparing the score-to-measure curves of the two DIF groups assesses whether DIF is large enough to cause differential functioning of the whole test (differential test functioning, actually [66]).

What is shown here is that questionnaire measures are robust to some amount of DIF. The measurement artefact caused by DIF at the item level is thinned in the questionnaire total score. However, it should be stressed that malfunctioning at the single item level remains. This fact should be considered if a subset of the questionnaire’s items is used to measure persons, such as in the computerized adaptive assessment. If most of the items selected for measuring are affected by DIF, the measurement artefact cannot be ignored anymore.

To our knowledge, only one other study compared the functioning of the Mini-BESTest items before and after rehabilitation [25] and, similarly to us, did not find DIF for treatment. However, this previous study only recruited subacute stroke patients. Despite its importance, DIF for treatment is seldom considered in psychometrics studies. This could be because it doubles the data collection effort and requires advanced analysis methods to deal with repeated measures.

#### 4.1.3. Assessing Construct Validity with the Item Map

Two critical threats to validity can be recognized: construct-irrelevant variance and construct under-representation [67].

Construct-irrelevant variance means that the questionnaire scores are contaminated by variables unrelated to the variable the questionnaire aims to measure. Tests of data-model fit, especially the PCA of the model’s residuals and the DIF analysis discussed above, examine if there is any construct-irrelevant variance.

Construct under-representation, i.e., when the assessment fails to include some essential facets of the construct, can be quickly assessed with the item map. For example, a wide gap between two neighboring items (or thresholds) in the item map points out that a range of the latent variable is not accessed by the questionnaire and is eventually poorly measured [21].

The item map of the Mini-BESTest showed that the distance between consecutive thresholds is narrow overall, indicating that the balance continuum is neatly probed by its items. However, it must be acknowledged that there are a few large gaps at the lower end of the map, a finding indicating some construct under-representation of low balance levels. Nevertheless, it is unlikely that this represents a critical weakness of the scale. First, the total number of major gaps in the threshold maps is low, just 3 out of 27. Second, very few persons had such a low balance level, a finding immediately apparent when the thresholds and the person maps are compared.

Finally, with the item map, it can be verified whether the item hierarchy along the latent variable line (e.g., here, the progression from easy to challenging balance tasks) is in line with what is expected given the theory about the latent variable under study [20]. According to the construct validity definition, the item map allows hypothesis testing about the items given the available knowledge about the construct.

For example, regarding static balance, the analysis reported here shows that it is easier to stand with feet together with the eyes open on a firm surface (item 7) than standing with the eyes closed on an inclined support (item 9). In turn, item 9 is much easier than standing with feet together with the eyes closed on an unstable surface (item 8). Finally, standing on one leg (item 3) is the most difficult static balance task. These findings are in full accordance with the common clinical experience and previous experimental evidence (e.g., [13,22,23,25,68]).

The Mini-BESTest maps reported in the current study also include the session map, highlighting that the discharge session was 0.88 logits easier than the admission one. This shift in the session’s difficulty, due to the clinical improvement, is of major importance for planning the sample size of trials using the Mini-BESTest as the primary outcome.

### 4.2. Study Limitations and Future Developments

Three principal limitations of the present work are worth mentioning.

First, the study sample size could be considered too small for a DIF analysis. Recent guidelines for the Rasch analysis indicate that at least 200 participants per group are needed for studying DIF [69,70]. However, not everyone agrees with this recommendation [47], which is taken directly from the educational literature and transferred “as it is” to the medical field. As for any statistical testing, also in the Rasch analysis, the model parameters’ reliability is given by their standard errors. Regarding the DIF examination reported here, these standard errors suggest that estimates were sufficiently precise (Table 4).

The second limitation concerns the assessment of DIF for assistive devices. In the current analysis, the examiner noted whether the patient used an assistive device during testing. Future studies could also consider the specific items in which the assistive device was used. Items completed with an assistive device could be scored without applying the “remove 1 point” correction and entered in the database alongside the items completed without assistive devices. Item calibrations would eventually be available for comparing balance in patients who used an assistive device in some specific items with those who completed the test without assistive devices or using an assistive device in a different set of items.

Finally, DIF for diagnosis can be further tested in other clinical groups, such as Parkinson’s disease and multiple sclerosis.

The current work showed that the Mini-BESTest has good metrological features in terms of reliability and validity. However, regarding the future developments of this line of research, other Mini-BESTest features such as its ability to predict future falls should be investigated further. Furthermore, it also seems natural to compare the functioning of this clinical balance measure with other balance measures, such as instrumental ones [7,8,9,10,11,12].

When compared to instrumental measures, clinical scales have advantages and disadvantages. Among the former, they use no technology; thus, costs and the level of expertise for data management and analysis are relatively low. On the other hand, however, more clinical expertise is needed to administer scales. For example, some competence in balance assessment is necessary to administer the Mini-BESTest. Instead, to cite an instrumental balance test, the instrumented TUG [10,11] could be easily collected by the general clinical staff (e.g., nurses) or even operators with no clinical background (e.g., caregivers for home mobility measurement).

Again, regarding the comparison between measurements, the Mini-BESTest psychometric features could also be compared to those of patient-reported outcome measures of the balance construct. The Mini-BESTest is a clinician-reported measure, a scale filled by a clinician (typically a physical therapist) who rates different patients’ behaviors.

Alongside clinician-reported, there are patient-reported outcome measures, which are questionnaires completed directly by patients [71]. Patient-reported outcome measures return measures of patients’ perceptions of their symptoms, activity limitations, and impact of the disease in their lives [71]. Most importantly, with self-administered questionnaires, patients are measured without the intermediation of another person: patients measure themselves. Patient-reported outcome measures have been encouraged, and it has been suggested that using them is associated with better health outcomes [72].

Patient-reported outcome measures of gait and balance constructs are available (e.g., the Falls Efficacy Scale International for measuring concern about falling [31]). Interestingly, these measures can have better psychometric features, such as higher responsiveness (e.g., [73]), than clinician-administered tests.

## 5. Conclusions

The Mini-BESTest returns balance measures with high reliability, strong construct validity, and robust measurement invariance. These can be safely used to compare balance in different conditions of clinical interest, such as before and after treatment and in different clinical groups. However, caution is needed when using the Mini-BESTest to compare balance in patients using an assistive device during testing.

## Figures and Tables

**Figure 1 ijerph-20-05166-f001:**
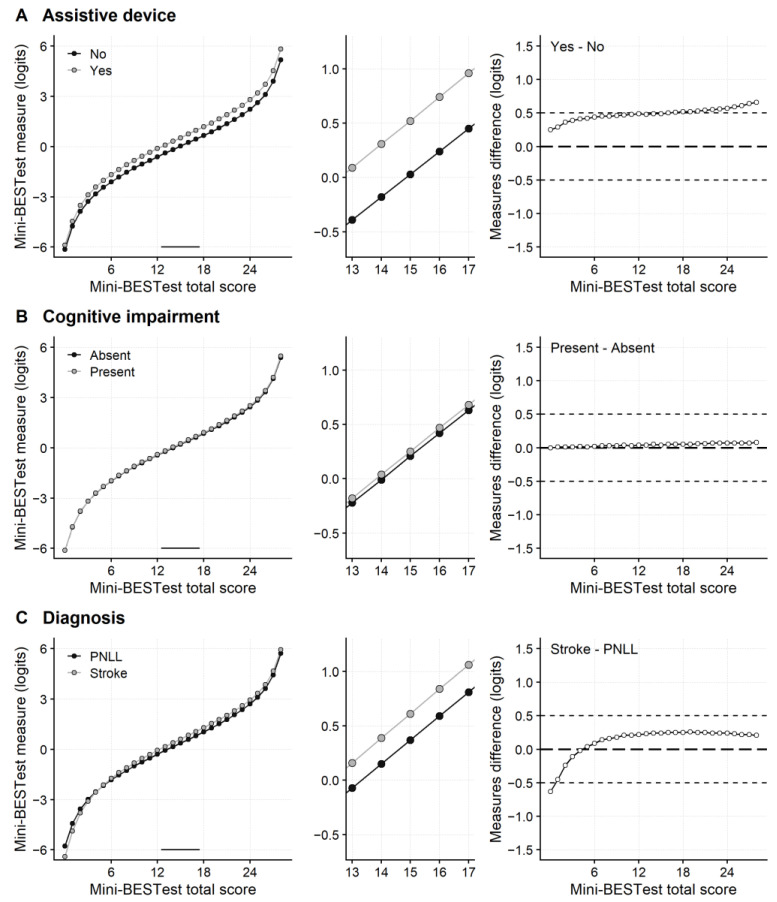
Differential Item Functioning of the Mini-BESTest scale. The artefact caused by the differential item functioning (DIF) on the Mini-BESTest measures has been studied by comparing the score-to-measure curves. DIF was found for assistive device use (**A**), cognitive impairment (**B**), and diagnosis (stroke vs. peripheral neuropathy of the lower limbs, PNLL) (**C**). The group-specific score-to-measure curves are shown in the leftmost plots. The middle graphs magnify the portion of the score-to-measure curves marked by a horizontal segment in the score-to-measure plots. The rightmost graphs show the difference between the two score-to-measure curves (*y*-axis) for each value of the Mini-BESTest total score (*x*-axis). This difference was about 0.5 logits (thin dashed horizontal lines) or larger for most of the Mini-BESTest scores in the case of DIF for assistive devices. In the case of DIF for diagnosis, the difference between the two score-to-measure curves was >0.5 only for a total score equal to 0.

**Figure 2 ijerph-20-05166-f002:**
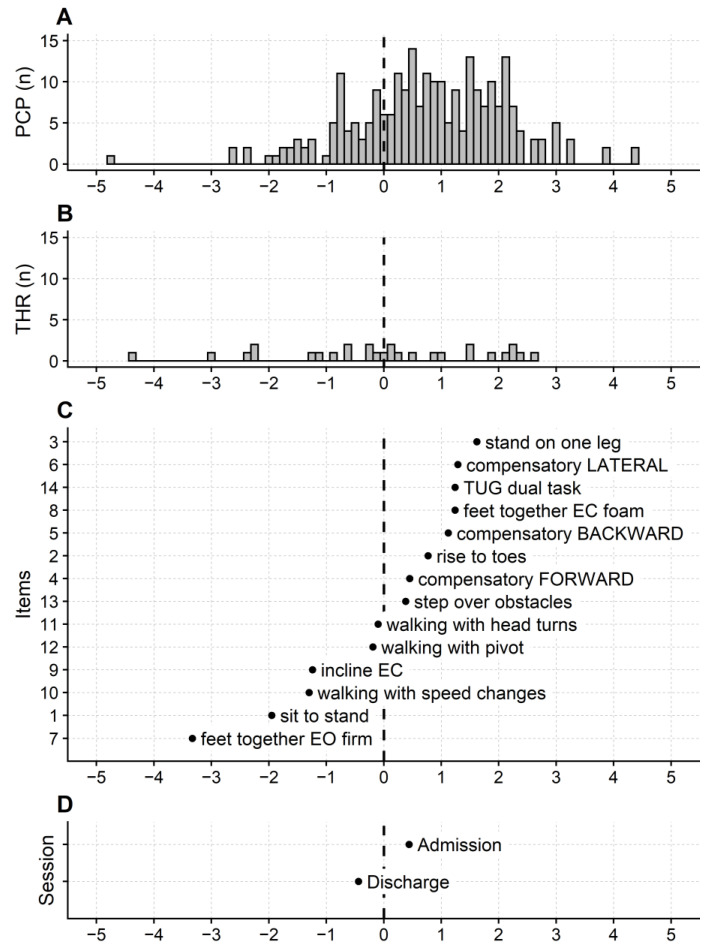
Maps of the Mini-BESTest scale. (**A**) distribution frequency of the patients’ measures. (**B**) distribution frequency of the Andrich thresholds. (**C**) item calibrations. (**D**) session calibrations (admission vs. discharge). Participants, items (with the Andrich thresholds), and sessions are the first, second, and third facets of the Many-Facet Rating Scale model used for the Rasch analysis. PCP: participants; THR: thresholds; n: number of. Item content is abbreviated with a keyword, and the ordinate gives the item number in the questionnaire. Items are ordered from low to high, from the easiest balance task to the most difficult to complete.

**Table 1 ijerph-20-05166-t001:** Participants’ clinical features.

gender, N females (%)		128 (43.8)
age, years, median (1st–3rd Q)		75.9 (67.7–81.3)
age class, N (%)	adults	59 (20.2)
	older adults	141 (48.3)
	very older adults	92 (31.5)
diagnosis, N (%)	Ischaemic stroke	119 (40.8)
	Haem. stroke	36 (12.3)
	PNLL	62 (21.2)
	Lumb. Rad.	24 (8.2)
	PD	23 (7.9)
	VP	28 (9.6)
cognitive impairment, N (%)	at T0 and T1	157 (53.8)
	at T0 but not T1	28 (9.6)
	at T1 but not T0	10 (3.4)
	neither at T0 nor T1	97 (33.2)
assistive device, N (%)	at T0 and T1	40 (13.7)
	at T0 but not T1	29 (9.9)
	at T1 but not T0	0 (0.0)
	neither at T0 nor T1	223 (76.4)
FIM-mot, total score, median (1st–3rd Q)	T0	66 (50–75)
	T1	80 (73.75–86)
FIM-cog, total score, median (1st–3rd Q)	T0	33 (29–35)
	T1	34 (30–35)
Mini-BESTest, total score, median (1st–3rd Q)	T0	14 (9–18)
	T1	18 (13–23)
gait speed, m/s, median (1st–3rd Q)	T0	0.72 (0.54–0.98)
	T1	0.92 (0.70–1.11)
TUG duration, s, median (1st–3rd Q)	T0	17.7 (13.2–25.5)
	T1	14.1 (11.1–18.7)

N: number of persons; Heam: haemorrhagic; PNLL: peripheral neuropathy of the lower limbs; Lumb. Rad.: lumbar radiculopathy; PD: Parkinson’s disease; VP: vascular parkinsonism; T0: admission assessment session; T1: discharge session; Q: quartile.

**Table 2 ijerph-20-05166-t002:** Mini-BESTest categories.

Categories	Count	%	Average Measures	Andrich Thresholds	
				Calibration	SE
0, severe	1335	20	−1.14	-	-
1, moderate	2472	37	0.24	−1.06	0.04
2, normal	2843	43	2.30	1.06	0.03

Categories: categories of the Mini-BESTest items (numeral and description). Count and %: number and percentage of observations of this category used in the analysis. Average measures: mean measures (logits) of the persons who chose the corresponding category. Calibration: measures (logits) of the Andrich thresholds (Many Facet Rating Scale model). SE: standard error (logits). The threshold between categories 0 and 1 is reported on the second row, and so on.

**Table 3 ijerph-20-05166-t003:** Mini-BESTest items calibrations and fit to the Rasch model.

			Infit		Outfit	
	Calibration	SE	MNSQ	ZSTD	MNSQ	ZSTD
1, sit to stand	−1.95	0.11	1.00	0.01	0.79	−1.30
2, rise to toes	0.77	0.08	0.93	−1.32	0.91	−1.26
3, stand on one leg	1.62	0.08	0.75	−4.71	0.81	−2.48
4, compensatory FORWARD	0.45	0.08	1.16	2.69	1.16	2.21
5, compensatory BACKWARD	1.12	0.08	1.11	1.83	1.07	0.97
6, compensatory LATERAL	1.29	0.08	1.13	2.17	1.09	1.17
7, feet together EO firm	−3.33	0.17	1.08	0.58	0.58	−1.40
8, feet together EC foam	1.24	0.08	1.30	4.71	1.22	2.83
9, incline EC	−1.24	0.10	1.29	3.75	1.04	0.34
10, walk with speed changes	−1.30	0.10	0.93	−1.02	0.75	−2.20
11, walk with head turns	−0.10	0.08	0.83	−3.06	0.87	−1.86
12, walk with pivot	−0.19	0.08	0.74	−4.69	0.77	−3.26
13, step over obstacles	0.38	0.08	0.87	−2.32	0.83	−2.64
14, TUG dual task	1.24	0.08	1.20	3.24	1.23	2.95

Each item is identified by its number and a keyword (leftmost column). SE: standard error; MNSQ: mean square; ZSTD: z-standardized; EO: eyes open; EC: eyes closed. Calibrations and SE are reported in logits.

**Table 4 ijerph-20-05166-t004:** Differential Item Functioning of the Mini-BESTest items.

	**Assistive Device**						
	**Tested with**		**Tested without**				
	**Calibration**	**SE**	**Calibration**	**SE**	**Contrast**	**SE**	***p*-Value**
1, sit to stand	−1.30	0.17	−1.91	0.11	0.60	0.20	0.003
4, compensatory FORWARD	1.04	0.21	0.40	0.08	0.64	0.22	0.005
6, compensatory LATERAL	1.96	0.27	1.08	0.08	0.87	0.28	0.003
10, walk with speed changes	−0.72	0.17	−1.27	0.10	0.55	0.19	0.005
12, walk with pivot	0.49	0.19	−0.21	0.08	0.70	0.21	0.001
13, step over obstacles	2.31	0.32	0.33	0.08	1.98	0.32	<0.001
	**Cognitive impairment**						
	**Present**		**Absent**				
	**Calibration**	**SE**	**Calibration**	**SE**	**Contrast**	**SE**	** *p* ** **-Value**
14, TUG dual task	1.47	0.11	0.94	0.12	0.53	0.16	0.001
	**Diagnosis**						
	**Stroke**		**PNLL**				
	**Calibration**	**SE**	**Calibration**	**SE**	**Contrast**	**SE**	** *p* ** **-Value**
7, feet together EO firm	−4.06	0.34	−2.83	0.30	−1.23	0.46	0.007
11, walk with head turns	0.23	0.11	−0.60	0.19	0.83	0.22	<0.001
12, walk with pivot	0.04	0.11	−0.64	0.19	0.69	0.22	0.003
14, TUG dual task	1.69	0.11	0.38	0.18	1.31	0.21	<0.001

The items affected by differential item functioning (DIF) for assistive device use (top), cognitive impairment (middle), and diagnosis (bottom) are reported in the leftmost column. Calibration: item calibration in the two groups of the DIF analysis; SE: standard error; Contrast: the difference between the item calibration in the two DIF groups; *p*-value: DIF significance from a t-statistic; Stroke: patients affected by hemiparesis due to a stroke; PNLL: peripheral neuropathy of the lower limb; EO: eyes open. Only the items with a large and significant DIF are reported. No DIF was found for age, gender, and session.

**Table 5 ijerph-20-05166-t005:** Score-to-measure conversion of the Mini-BESTest.

Total Score	Measure (Logits)	SE (Logits)
0	−6.04	1.90
1	−4.65	1.12
2	−3.71	0.85
3	−3.10	0.73
4	−2.63	0.65
5	−2.23	0.60
6	−1.89	0.57
7	−1.58	0.54
8	−1.30	0.52
9	−1.04	0.50
10	−0.80	0.49
11	−0.56	0.48
12	−0.34	0.47
13	−0.12	0.47
14	0.10	0.46
15	0.31	0.46
16	0.52	0.46
17	0.74	0.47
18	0.96	0.47
19	1.19	0.48
20	1.42	0.49
21	1.67	0.51
22	1.93	0.53
23	2.22	0.55
24	2.55	0.59
25	2.94	0.66
26	3.45	0.77
27	4.24	1.05
28	5.51	1.85

SE: standard error.

## Data Availability

The data presented in this study will be available on Zenodo upon acceptance of the paper.

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
