# Peer review of "Differential Item Functioning of the Mini-BESTest Balance Measure: A Rasch Analysis Study"

_ijerph, 2023, doi:10.3390/ijerph20065166_

Round 1
Reviewer 1 Report
Thank you for the opportunity to review the manuscript "Differential Item Functioning of the Mini-BESTest balance measure: a Rasch analysis study." by Caronni et al.
Differential Item Functioning is important for measurements because it helps identify whether an item in the Mini-BESTest behaves differently for different subgroups of individuals, even when they have the same ability or trait being measured. This is important because if an item has differential functioning, it can introduce bias into the measurement and affect the validity and fairness of the Mini-BESTest. By detecting and addressing DIF, the authors can improve the accuracy and fairness of the instrument.
However, there are some points that I would like to see considered in a revision:
1) Abstract: In the abstract, only little is said about the background and why the study is being done. Since Rasch-statistics is not familiar to every reader, this would be good to increase readability.
2) The authors write: " The theoretical underpinnings of the Rasch analysis can be found in [14,32,33], while the technical details of the analysis run here can be found in [26,34,35]." However, I believe that this is not sufficient. Please provide more details either in the manuscript or in a supplement. Ideally, one can read and understand this study without searching and reading other studies.
3) In the introduction, I would also like to see more about the results from the Rasch analyses, like assumptions of local independence and unidimensionality.
4) Method: On what basis were the variables studied selected?
5) I did not know the "Mini-BESTest balance measure" before reading this manuscript. In general I find it a pity that the test and also the other assessments used are purely "Professionals-reported-Measures" and do not include the patients' perspective (PRO(M)s). For example, self-perceived safety/balance could perhaps also play a role here? Can this be added to the discussion?
6) What really bothers me is that the authors use the term "elderly", which is actually a term that should no longer be used. Elderly is problematic because of its association with dependency and frailty, which typecasts whole generations. Although it is natural that we refer to people in groups by their age, chronological age is a poor guide to understand any given person; their health, their personality and their experience. Please use instead "older adults" or similar.
Reviewer 2 Report
Thank you for your manuscript. Your topic is interesting and provides additional insight into balance assessment tools. Please see my comments below…
ABSTRACT
P1L13. “The Mini-Balance Evaluation Systems Test (Mini-BESTest), a 14 items scale, has high content validity for balance assessment.” – During the Introduction I cannot find references that support this sentence. Moreover: “Evaluations of content validity include systematic comparison of a measure by confronting it with existing standards, widely accepted theoretical definitions, expert opinions, and interviews with individuals to whom the measure is targeted.” Krabbe P. The Measurement of Health and Health Status. Concepts, Methods and Applications from a Multidisciplinary Perspective. London: Elsevier; 2017.
INTRODUCTION
P1L43. “Many balance measures are available” [7–10]. Which? Moreover, I believe that some references do not refer to balance measures. First, I believe TUG itself does not assess balance. Second, I believe that TUG’ convergent validity regarding balance must use a gold standard tool and not scales as comparable methods. If Mini-BESTest were a gold standard and validate tool the present study did not make sense. Although I believe that some biomechanical parameters can be used as that.
P1L43. “Among the balance scales…”. I believe that is needed to address advantages regarding scales for assessing balance.
P1L43. “First, since their ordinal scores can be turned into interval measures, they provide measures like…” Did you mean “continuos variables” and “variables” instead of “interval measures” and “measures”?
P2L54. “…a questionnaire passing the Rasch analysis has good construct validity [15,16].” Rasch analysis assesses convergent or divergent validity? “Construct validity is now generally viewed as a unifying form of validity for measurements. It subsumes both content and criterion validity, which traditionally had been treated as distinct forms. (…) “Convergent and discriminant validities are two fundamental aspects of construct validity. Convergent validity refers to how closely the new scale is related to other variables and other measures of the same construct. Not only should the construct correlate with related variables but it should not correlate with dissimilar, unrelated ones.” Krabbe P. The Measurement of Health and Health Status. Concepts, Methods and Applications from a Multidisciplinary Perspective. London: Elsevier; 2017.
This comment will have implications for other sentences in the text.
METHODS
P3L98. “Participants were included if: (…)iii. Able to walk without touching assistance and.” This sentence is not coherent with this: P3L127. “However, if required, they were allowed to use their assistive device (most commonly a walker) for one or more items.
P3L111. Is not clear if the assessments were made during one only session and the respective order of the tests.
P5L220. “Patients scoring < 35 (out of 35) on the cognitive domain of the FIM scale were considered cognitively impaired. Conversely, patients scoring 35/35 were supposed to suffer no cognitive impairment.” Why? Reference?
Round 2
Reviewer 1 Report
The authors did a good job of incorporating my comments and feedback. I wish the authors all the best for the publication.